# Practical Relevance of Institutional Guidelines in Translational Large Animal Studies of Cartilage Repair—A Multidisciplinary Survey

**DOI:** 10.3390/medicina58121834

**Published:** 2022-12-13

**Authors:** Moritz Riedl, Markus Rupp, Nike Walter, Leopold Henssler, Maximilian Kerschbaum, Daniel Popp, Gianluca Vadalà, Volker Alt, Denitsa Docheva, Christian G. Pfeifer

**Affiliations:** 1Department of Trauma Surgery, University Regensburg Medical Centre, 93053 Regensburg, Germany; 2Laboratory of Experimental Trauma Surgery, University Regensburg Medical Centre, 93053 Regensburg, Germany; 3Department of Orthopaedic and Trauma Surgery, Campus Bio-Medico, University of Rome, 00128 Rome, Italy; 4Department of Musculoskeletal Tissue Regeneration, Orthopaedic Hospital König-Ludwig-Haus, University of Wuerzburg, 97070 Wuerzburg, Germany

**Keywords:** cartilage repair, translational study, survey, guidelines, animal model

## Abstract

*Background and Objective*: Translational large animal models are inevitable to transfer cartilage repair methods into clinical practice. Guidelines for these trials have been published by guiding agencies (FDA, ASTM, EMEA) including recommendations for study descriptors and study outcomes. However, practical adherence to these recommendations is not achieved in all aspects. This study includes an assessment of the recommended aspects regarding practical relevance in large animal models for cartilage repair by professionals in the field. *Materials and Methods*: In an online based survey, 11 aspects regarding study design and 13 aspects regarding study outcome from previously published guidelines were evaluated (0–10 points, with 10 being most important) by study participants. Additionally, the survey contained questions related to professional experience (years), professional focus (preclinical, clinical, veterinarian, industry) and the preferred translational large animal model for cartilage repair. *Results*: The total number of survey participants was 37. Rated as most important for study design parameters was lesion size (9.54 pts., SD 0.80) followed by study duration (9.43 pts., SD 1.21); and method of scaffold fixation (9.08 pts., SD 1.30) as well as depth of the lesion (9.03 pts., SD 1.77). The most important aspects of study outcome were considered histology (9.41 pts., SD 0.86) and defect filling (8.97 pts., SD 1.21), while gene expression was judged as the least important (6.11 pts., SD 2.46) outcome. A total of 62.2% of all participants were researchers, 18.9% clinicians, 13.5% veterinarians and 5.4% industry employees. *Conclusions*: In translational research, recommendations published by guiding agencies receive broad theoretical consensus within the community, including both clinically and preclinically orientated scientists. However, implementation into practical research lacks in major aspects. Ongoing re-evaluation of the guidelines under involvement of all stakeholders and approaches to overcome financial and infrastructural limitations could support the acceptance of the guidance documents and contribute to standardization in the field.

## 1. Introduction

Traumatic articular cartilage lesions are a growing medical issue as articular cartilage damage predisposes posttraumatic osteoarthritis, especially in an aging population. Untreated defects lead to further degeneration and severe disabling joint disease [1].

Due to limited regeneration capacity of hyaline cartilage in adult human joints, articular chondral lesions represent a therapeutic challenge in orthopedics. During recent decades, several approaches to articular cartilage repair have been established. Bone marrow stimulation [2], osteochondral transplantation [3] and autologous chondrocyte transplantation (ACT) [4] are the most commonly used and most discussed methods. Continuously, innovative therapy options are developed based on enhancements of these fundamental techniques [5]. However, the path from basic research towards clinical application in patients requires the preclinical evaluation in animal models. In general, large animal models (i.e., equine, ovine, caprine and porcine) serve as pivotal preclinical studies and allow for a critical assessment of treatment efficiency and safety. Alongside the choice of the right species, investigators have to consider whether their study design will convince the peer community as well as regulatory agencies and that well-performed pivotal trials need to produce data legitimating a clinical trial in humans. Especially in terms of animal welfare, high-quality trials are of certain importance. Gaining the maximum information out of a well-performed study can reduce the number of necessary animals. On behalf of the 3R principles for animal experiments (replacement, reduction, refinement) [6], animal trials in translational research should, if they cannot be avoided, at least follow the maxims of “reduction” and “refinement”, which in the end is a matter of quality. For orientation on that point, specific guidance documents have been published by governing agencies including the U.S. Food and Drug Administration (FDA) [7], the European Medicines Agency (EMA) [8] and the American Society for Testing and Materials (ASTM) [9]. These guidance documents feature assistance for proper execution of final preclinic studies including specific instructions with regards to study description and outcome parameters. However, as the guidance documents are only recommendations, the adherence is not mandatory. In view of the voluntary basis, Pfeifer et al. analyzed how these guidelines are adhered to within the community in the field of articular cartilage regeneration and to what extent they are followed in the design of correspondent trials [10]. They investigated the concordance of the guidance documents with the reality in studies published from 1994 to 2014 and described the impact of the single documents on following publications. Their results indicate a lack of standardization in the field. A weak positive trend of adherence over the period of 20 years could be observed, but only a maximum overall adherence of 58% to the guidelines was reached.

In this study, we aimed to evaluate how scientists in the field rate the importance and practical relevance of the aspects included in the published guidelines in order to identify reasonable recommendations on the one hand and aspects that lack in relevance or feasibility on the other hand. Therefore, we performed a survey among scientists working in the field of orthopedic clinical and basic research regarding the relevance of the single aspects listed in governmental guidance documents. We are interested in the question of whether the instructions outlined in the guidance documents generally meet with the expectations of the community, if their deficient realization is hence due to other reasons, or if the expectations of the agencies just do not agree with the real-world conditions in research. This study should serve as feedback to the responsible agencies. Ongoing improvement of available guidelines under participation of all stakeholders can raise quality and standardization in the important field of translational science.

## 2. Materials and Methods

In this cross-section study we performed an anonymized, web-based survey in collaboration with the European Orthopaedic Research Society (EORS). EORS is an international society with the main mission of promoting research and development in orthopedic surgery and related sciences in Europe through interdisciplinary coordination, exchange of scientific and technical experience, and education. The survey was offered to the EORS members and participants of the EORS Annual Meeting (2017) during the period of August 2018 to January 2019. Initial analysis of the guidance documents of FDA, EMA and ASTM identified 24 aspects that are claimed by the regulatory agencies as criteria of a well-performed pivotal trial. These are defined in Table 1, classified into study descriptors and study outcomes, according to the agencies standards.

The participants were asked to rate every single aspect on a scale from 1–10 regarding its importance, in which 10 represents the maximum importance. 

Furthermore, they were asked to set up a ranking order grading the criteria according to decreasing importance in their opinion. Additionally, general information about the respondent was obtained, regarding professional category, academic degree, work experience, research focus (clinical, basic research, translational) and preferred animal model and species.

The survey was performed via Google Forms (Figure 1) with one reminder over a period of six months from August 2018 to January 2019. Participation was anonymous besides the above requested data, which participants provided voluntarily.

Data are expressed as mean value ± standard deviation (SD). Additionally, the standard error of the mean (SEM), interquartile range (IQR) and the 95% confidence interval were calculated. The Kolmogorov Smirnov test was used to evaluate data for equal distribution and the *t*-test was used for statistical analysis and *p* < 0.05 was considered significant. 

## 3. Results

During the study duration, 37 researchers participated in the survey. More than half of the participants claimed more than 10 years of experience in the field of articular cartilage regeneration. The study participants represented mainly academic scientists/engineers (23/62.2%) and clinicians (i.e., orthopedic surgeons) (7/8.9%) while five veterinarians (5/13.5%) and two members of industrial companies also completed the survey. The overall academic experience was high, as 70.3% (26 of 37) of the respondents are lead researchers (e.g., principal investigators).

The evaluation of the results showed high consensus between researchers and agencies regarding the study descriptors. Only animal weight and gender seem to be less important for the participants with ratings of 7.57 ± 2.30 and 6.92 ± 2.75 points (max. 10), respectively. All other criteria ranked from 8.35 ± 1.89 to 9.54 ± 0.80 points. The best ranked criteria are defect size, study duration and animal age on positions 1–3, thus indicating maximum importance.

Regarding the study outcome parameters, the investigators did not agree with the guidelines proposed by the guiding agencies. Only histological analysis received a nearly concordant appreciation, with a score of 9.41 ± 0.86 points. Parameters describing the defect site including gross view, defect fill, integration and evaluation of adjacent cartilage and subchondral bone got general acceptance. Their ratings ranged from 8.08 ± 1.80 to 8.97 ± 1.21 points. More experimental analyses such as gene expression (6.11 ± 2.46 points) and biochemistry (7.62 ± 2.01 points) as well as follow-up examinations via arthroscopy and MRI (6.49 ± 2.08, respectively, 7.51 ± 1.97 points) were less relevant in eyes of the interviewees. By comparison, outcomes more closely related to clinical application, such as biomechanical (8.40 ± 1.96 points) and clinical evaluation (8.11 ± 1.76 points), seem to be favored. According to the ranking histology, integration into the adjacent cartilage and grade of defect filling are the most important outcomes for the participating researchers (Figure 2).

Figure 3 illustrates the survey scores for every single criterion compared to the remaining guideline aspects. 

Alongside the overall results, we also faced the different outcomes between certain groups of investigators. We were interested in the different opinions of researchers relatively new in the field (<10 years) and those more experienced (>10 years). In general, the group of experienced scientists assessed the relevance of the guideline criteria higher than the group with less work experience. In the well-versed group, we found statistically significant higher values for describing factors including lesion size (*p* = 0.031) and depth (*p* = 0.001), study duration (*p* = 0.003), animal age (*p* = 0.009) and rehabilitation protocols (*p* = 0.021) and for the outcome related parameters, gross view (*p* = 0.021) and biomechanical testing (*p* = 0.017), compared with the less experienced cohort. We also made a second comparison according to the investigator’s profession between more basic scientific participants (scientists, employees in pharmaceutical industry) and clinical working researchers including physicians and veterinarians. The study descriptors lesion location (*p* = 0.030) and depth (*p* = 0.001), as well as animal age (*p* = 0.034), were significantly more important for the clinical group compared to the basic research group (Figure 4).

## 4. Discussion

Standardization and persistent quality are general aims in translational research. Guidance documents for translational intended large animal studies, especially in the field of cartilage regeneration, are published by governing agencies as control structures for investigators to meet that target and to pave the way from preclinical investigations to clinical applications. In cartilage regeneration, research guiding agencies, e.g., FDA, ASTM, EMA, deliver guidelines related to study design and description of pivotal large animal trials. However, the adherence to those guidelines in practical translational research is not as high as desired. We performed a survey among orthopedic and musculoskeletal scientists to evaluate the acceptance of single categories published within these guidance documents and to find reasons for their relatively low incorporation in translational study designs of cartilage repair studies. 

A total of 24 guideline categories were identified and rated by the participants according to their subjectively judged importance. Apart from single aspects, the overall rating was relatively high. Seventeen criteria reached survey values of more than eight out of ten points. The study descriptors especially received wide consensus with a mean score of 8.66 points. Categories related to study outcome still reached a mean value of 7.98. Pfeifer et al. reported that study descriptors were in general reported by ~75% of the analyzed publications. However, among study outcomes, only three criteria (histology, gross view and grade of defect fill) met a rate of adherence of more than 50% [10]. 

Thus, the question arises as to how this discrepancy between general approval and deficient realization can be explained. 

As one can see, study descriptors are not the main problem. The sufficient study description is likely a matter of standardization and accuracy. Improvements could be achieved by reporting more detailed information such as age, weight, and gender of study animals. These categories are lowly rated parameters in our survey as well as underrepresented study descriptors in current publications, but they can have a major impact on cartilage repair potential in humans [11,12,13]. If this data is collected and documented from the beginning of a trial, it does not mean an extra effort to include them in the description is needed. Data can be visualized in comprehensive elements, e.g., flow charts and tables, for easily accessible information. However, study description should already be part of the prior planning to avoid loss of information. Therefore, the ARRIVE (Animals in Research: Reporting In Vivo Experiments) guidelines [14] provide a compact yet comprehensive manual for the publication of animal studies and guarantee a sufficiently detailed description of in vivo trials in order to make them reproducible and transparent for researchers in the field. 

However, while study descriptors are generally at an acceptable level and improvements are easily feasible, study outcomes are lacking in adherence across the field. Most points might meet consensus in theory and for sure should be aspired to, but the implementation fails due to financial, organizational, or technical reasons. Expectedly, outcome measures that are self-evident and less elaborate, e.g., gross view, defect filling and histology, are prevalent in translational studies and rated highly in our survey as well, whereas those requiring specialized expertise (e.g., mechanical testing) are underrepresented even if they are highly rated in our survey. 

Others do not even achieve theoretical acceptance. Although the FDA explicitly recommends follow-up arthroscopy as an important diagnostic instrument [7], this aspect did neither meet approval in our cohort nor is adhered to in recent translational large animal studies of cartilage repair [10]. Without any doubt, follow-up arthroscopy would provide useful insights in healing progress and allow for histological sample recovery at different time points. Arthroscopic procedures are clinically established in different joints of horses and dogs [15,16], while arthroscopic surgical treatment in pigs, sheep and goats is mainly pursued in experimental studies so far [17,18]. Thus, only few institutions looked into follow-up arthroscopy in sheep, pigs, goats and dogs, while others simply lack equipment. 

In order to overcome infrastructural problems, research collaborations and networks or even federally funded core facilities could be an approach to allow for general access to more advanced and therefore complete outcome measurements.

Furthermore, one has to discriminate between the intention of translational studies from pilot studies. A pilot study is a short-term and small-sample size study for exploratory and preparatory purpose. Its function is the proof of feasibility and to generate data for decision-making on whether a larger more elaborate study is warranted. Very often, a pilot study is underpowered to draw final conclusions. Consequently, pilot studies focus on only few outcomes to be proceeded in future studies. Pfeifer et al. included both kinds of studies into their evaluation as the above-mentioned guidance documents do not distinguish between true translational and pilot intention. However, not every study has the same intent and thus, does not have to meet the same criteria. Nevertheless, every study employing animal models has to follow ethical guidelines including the 3 Rs (replacement, reduction, refinement) [6,19]. On behalf of animal welfare, guidelines should intend to improve quality of translational research by appropriate yet realistic requests.

In some aspects, there are relevant discrepancies between clinical and basic research. Lesion location and depth, as well as animal age, are rated significantly higher by clinical researchers compared to basic researchers. This might reflect the focus of clinicians on aspects with direct practical relevance in cartilage therapy as patient age as well as location and depth of chondral lesions significantly dictate the choice of treatment in clinical practice. However, there is also scientific evidence for the importance of these parameters. e.g., increasing age correlates with less cartilage hydration and lower chondrocyte number and less proliferation capability [20,21].

Thus, it is crucial to respect the different approaches of clinically and preclinically orientated researchers likewise in these guidelines. 

Are the expectations of the guiding agencies too high for practical research?

According to the findings of our survey among researchers in the field of orthopedic and musculoskeletal regeneration, we think that the guidance documents are quite convincing, and the suggested outcomes are qualified and will gain acceptance if they are more routinely used. However, ongoing re-evaluation of the current documents will provide a set of recommended outcome measurements that is agreed upon (and followed) by all stakeholders and completed by an additional list of optional yet desirable outcomes. Additionally, guidance documents should discriminate between pivotal and pilot studies and be complemented by an extra set of criteria for pilot studies. As a consequence, authors of large animal studies should also clearly describe the intent of the published work as pivotal or translational study. Journals, founders, and institutes can contribute to standardization in the field and support the acceptance of the guidance documents by including them into their terms and conditions.

Collaborations, research networks and federally funded core facilities can help laboratories to overcome the lack of infrastructure and expertise needed for elaborate outcome methods and the pendency from commercial funding.

## 5. Conclusions

In translational research recommendations published by guiding agencies (FDA, EMA, ASTM) receive broad theoretical consensus within the community including clinically as well as preclinically orientated scientists. However, implementation into practical research lacks in major aspects. Ongoing re-evaluation of the guidelines with the involvement of all stakeholders (guiding agencies, scientists, clinicians, research institutes) and approaches (clinical, scientific) to overcome financial and infrastructural limitations could support the acceptance of the guidance documents and contribute to standardization, quality, and animal welfare in the field.

## Figures and Tables

**Figure 1 medicina-58-01834-f001:**
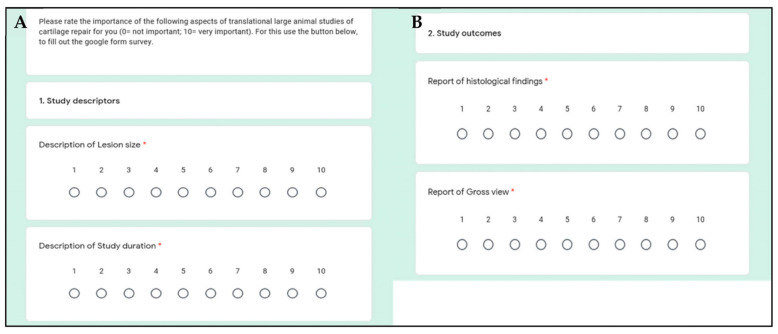
**Example of the internet-based survey.** (**A**) First two survey items regarding study descriptors (**B**) first two survey items regarding study outcomes.

**Figure 2 medicina-58-01834-f002:**
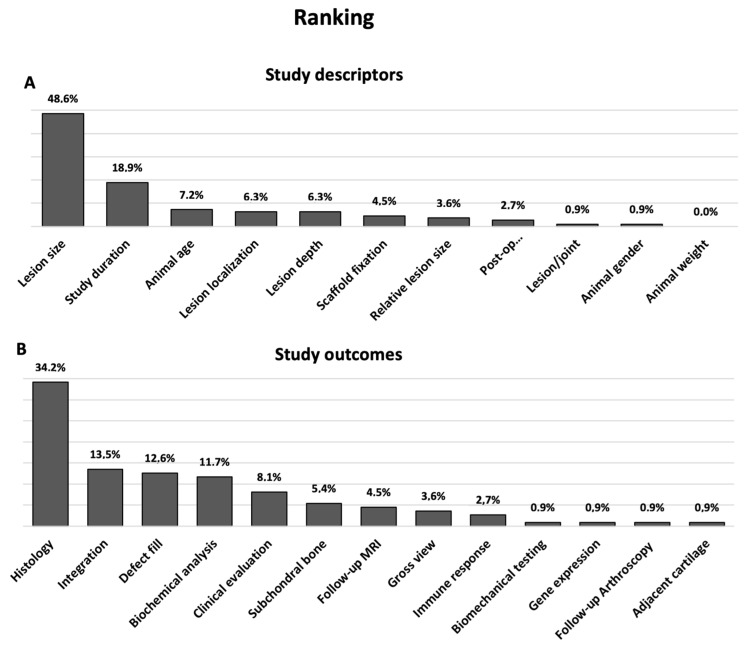
Ranking order of guidance documents criteria. Divided into (**A**) study descriptors and (**B**) study outcomes and arranged according to decreasing importance. The percentage represents the part of participants ranking the respective criteria as most important. *n* = 37 (participants).

**Figure 3 medicina-58-01834-f003:**
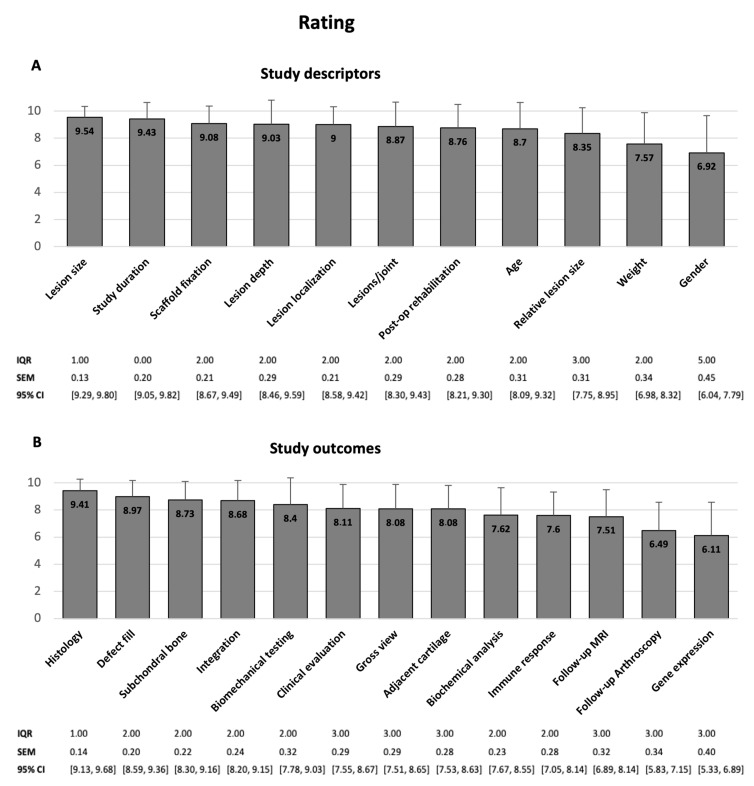
Illustration of the survey ratings for every single guideline aspect regarding to their individual importance in the design of pivotal animal trials. (**A**) study descriptors and (**B**) study outcomes, arranged according to decreasing importance. Whiskers represent standard deviations. *n* = 37 (participants); 10 = most important; IQR interquartile range; SEM standard error of the mean; CI confidence interval.

**Figure 4 medicina-58-01834-f004:**
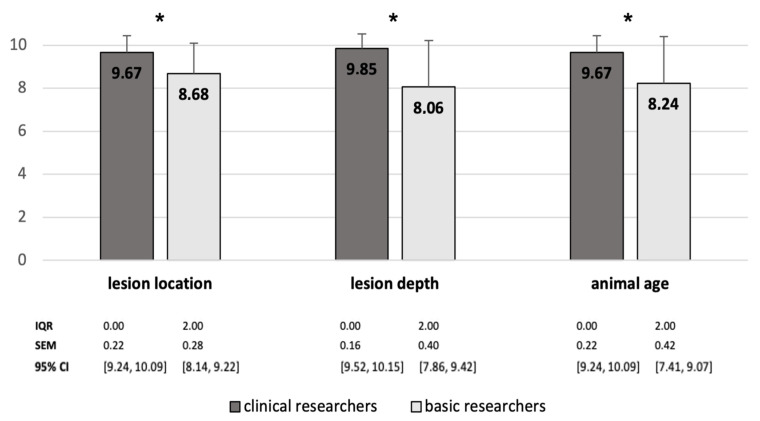
Survey aspects with statistically significant different ratings by clinical working researchers and basic researchers. Whiskers represent standard deviations. Significant differences (* *p* < 0.05) are designated with asterisks. IQR interquartile range; SEM standard error of the mean; CI confidence interval.

**Table 1 medicina-58-01834-t001:** List of 24 major aspects defined by the guidance documents.

Study Descriptors	Study Outcomes
Lesion size	Histology
Lesion localization	Gross view
Lesion depth	Biomechanical testing
Lesions/joint	Biochemical analysis
Relative lesion size	Gene expression
Study duration	Clinical evaluation
Post-op rehabilitation	Follow-up Arthroscopy
Scaffold fixation	Follow-up MRI
Age	Defect fill
Weight	Integration
Gender	Adjacent cartilage
	Subchondral bone
	Immune response

## Data Availability

The datasets used and analysed during the current study are available from the corresponding author on reasonable request.

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
