# Peer review of "Practical Relevance of Institutional Guidelines in Translational Large Animal Studies of Cartilage Repair—A Multidisciplinary Survey"

_medicina, 2022, doi:10.3390/medicina58121834_

Round 1

Reviewer 1 Report

This is an interesting look at the basic science studies looking at cartilage repair in animal models.

The authors should clarify their purpose-it is still a bit vague- and it remains unclear if the results will have any impact on practice.

Author Response

Dear Reviewer,

Thank you very much for your comments and evaluation of the presented work. We gratefully included your suggestion into the revision of our article and further pointed out the purpose of our study in the introduction section (please see lines 91-102).

Reviewer 2 Report

Dear authors, 

thank you for the interesting manuscript. I appreciate the insight your study provides and also the difficulty to collect such data, and yet number of participants is rather limited. Nonetheless, the study has its merits and is worth publishing as the discussion addresses important issues for the community. The data presentation is unfavorable as some statistics are missing in the manuscript. Please provide the descriptive statistics (IQR). Also please use boxplots instead of barplots with SDs as they are more informative. Indicate significant results with brackets and asterisk inside the plots.  Alternatively you could provide SEMs and confidence intervals. 

Author Response

Dear Reviewer,

Thank you very much for your comments and evaluation of the presented work. We gratefully included your suggestions into the revision of our article. All changes to the manuscript text had been made using the “track changes” function.

Following your recommendation, we revised the data presentation. As we think that the presented date is not best suited for a boxplot presentation we instead added the missing information including IQR, SEM and 95% CI to the existing figures.

Statistically significant differences were indicated by asterisks and p-values were added to the text (please see line 188-195).

Thank you very much for these improvements!